# The Relation of the Iron Metabolism Index to the Vulnerability Index of Carotid Plaque with Different Degrees of Stenosis

**DOI:** 10.3390/diagnostics13203196

**Published:** 2023-10-12

**Authors:** Wanzhong Yuan, Ran Huo, Chaofan Hou, Wenbin Bai, Jun Yang, Tao Wang

**Affiliations:** 1Department of Neurosurgery, Peking University Third Hospital, Beijing 100191, China; ywz@stu.pku.edu.cn (W.Y.); hcfky@outlook.com (C.H.); baiwenbinbjmu@163.com (W.B.); 13901291211@163.com (J.Y.); 2Department of Radiology, Peking University Third Hospital, Beijing 100191, China; ransmile216@163.com

**Keywords:** carotid plaque, vulnerable plaque traits, plaque morphology, histopathology, ferritin, transferrin receptor

## Abstract

Objective: To investigate the differences in serum iron index and iron metabolizing protein expression in plaques in patients with different degrees of carotid artery stenosis and the relationship with plaque traits. Methods: A total of 100 patients eligible for carotid endarterectomy (CEA) from August 2021 to February 2022 were included. Patients completed a computed tomography (CTA) scan for patient grouping and a magnetic resonance imaging (MRI) for precise quantification of carotid plaque traits within 1 week prior to surgery. Clinical indicators associated with the progression of carotid stenosis to occlusion were analyzed using ordered logistic regression. Twenty carotid plaques were analyzed immunohistochemically to investigate the relationship between plaque traits and the iron metabolism indexes. Results: No significant correlation between high serum ferritin (SF), unsaturated iron binding capacity (UIBC) and progression of carotid stenosis (OR 1.100, 95% CI 0.004–0.165, *p* = 0.039; OR 1.050, 95% CI 0.005–0.094, *p* = 0.031). SF and serum transferrin receptor (sTfR) were correlated with normalized wall index (NWI) (R = 0.470, *p* = 0.036; R = 0.449, *p* = 0.046), and the results of multiple linear regression suggested that SF and sTfR remained associated with NWI (R = 0.630, R^2^ = 0.397, Adjusted R^2^ = 0.326, *p* = 0.014). In plaques, H-type ferritin (H-FT) was correlated with NWI and lipid-rich necrotic core (LRNC) volume (R = 0.502, *p* = 0.028; R = 0.468, *p* = 0.043). Transferrin receptor 1 (TfR1) was correlated with LRNC volume and intraplaque hemorrhage (IPH) volume (R = 0.538, *p* = 0.017; R = 0.707, *p* = 0.001). Conclusions: There were statistical differences in the expression of iron metabolism proteins in carotid plaques with different degrees of stenosis. Serum iron metabolism index (SF and sTfR) and expression of iron metabolizing proteins (H-FT and TfR1) in plaques were positively correlated with carotid plaque vulnerability index (NWI, LRNC volume).

## 1. Introduction

Ischemic stroke is a major threat to health worldwide [1,2]. Carotid atherosclerosis (CAS) is the most common cause of ischemic stroke. Currently, CT angiography (CTA) and high-resolution magnetic resonance imaging (hr-MRI) are used to assess the degree of carotid artery stenosis and determine the vulnerability of carotid plaque and predict the severity of stroke [3,4,5].

Vulnerable plaques (VP) often have some specific traits: morphological features, such as high grade stenosis rate, plaque high volume load, etc., and/or compositional features, such as intraplaque hemorrhage (IPH), thrombus, lipid-rich necrotic core (LRNC), neovascularization, fibrous cap rupture, etc., that make the plaque more prone to dislodge and cause stroke [6].

Iron is not associated with the initiation of atherosclerosis; however, iron and its associated metabolic proteins are associated with the progression of atherosclerosis. In humans, ferritin can be classified into H-type ferritin (H-FT) and L-type ferritin (L-FT) depending on the subunit composition [7]. H-FT has ferroxidase activity, which is important for iron incorporation and inhibition of ferrous ion toxicity, thereby reducing oxidative damage [8]. L-FT may promote the progression of atherosclerosis by regulating lipid oxidation within the vessel wall through the generation of reactive oxygen species [9]. Transferrin receptor1 (TfR1) is commonly expressed on the cell surface, and interacts with transferrin to participate in cellular iron uptake, and serum transferrin receptor (sTfR) is more sensitive and reliable than SF [10,11,12]. In clinical studies, elevated serum ferritin (SF) can be a risk factor for the progression of carotid and coronary artery atherosclerosis [2,12,13,14]. However, the differences in serum iron indices and the expression and types of iron metabolism-related proteins in plaques of patients with different degrees of carotid stenosis require further study.

In the present study, we aimed to investigate (1) whether there are differences in serum iron metabolism indexes in patients with different degrees of carotid artery stenosis; (2) whether there are differences in the expression of iron metabolizing proteins in plaques in patients with different degrees of carotid artery stenosis; and (3) whether serum iron metabolism indexes, and iron metabolizing proteins expression in plaques, are correlated with carotid artery plaque vulnerability indexes.

## 2. Materials and Methods

### 2.1. Study Sample

A total of 100 patients who underwent CEA between August 2021 and February 2022 at our center were included. All patients underwent CTA and hr-MRI within 1 week prior to CEA. Demographic and clinical characteristics and laboratory information were collected from clinical records. The included patients underwent routine blood tests before CEA (including blood routine, blood biochemistry, iron metabolism indicators, etc.), and the samples were sent to the clinical laboratory of the hospital for standardized testing. Specific methods for measuring indicators of serum iron metabolism are shown in the Appendix A. In this study, the ranges of the four serum iron metabolism indices were as follows: SF 10.6–36.7 µmol/L; sTfR 20.8–59.16 nmol/L; total iron binding capacity (TIBC) 46.4–69.6 µmol/L; unsaturated iron-binding capacity (UIBC) 31–48 µmol/L.

All patients were free from the following diseases in their past history (in order to exclude the effect of ferritin increase in the acute phase): liver diseases, such as viral hepatitis and fatty liver; all neoplastic diseases; acute inflammation of organs and tissues of the organism; autoimmune diseases; all hematological diseases; and treatment with blood transfusions or iron supplements. All patients took aspirin (daily doses of 100 mg, oral) before CEA and stopped taking clopidogrel 1 week before CEA. Intraoperatively, intravenous heparin (single dose of 5000 µ) was routinely given before carotid artery clamping. After CEA, all patients were given heparin (2500 IU, within 24 h), aspirin (daily dose of 100 mg, oral), and intensive statin therapy (atorvastatin, daily dose of 40 mg, oral).

Written informed consent was obtained from each patient included in the study. The study protocol conforms to the ethical guidelines of the 1975 Declaration of Helsinki and the study protocol has been priorly approved by the Institution’s ethics committee on research on humans.

### 2.2. Imaging and Analysis of CTA and hr-MRI

The degree of stenosis of the patient’s carotid artery was calculated according to the MRC European Carotid Surgery Trial (ECST) method based on the patient’s CTA images. The 100 patients were categorized into five groups according to the degree of stenosis: group A (stenosis rate 60–70%), group B (stenosis rate 70–80%), group C (stenosis rate 80–90%), group D (stenosis rate 90–95%), and group E (stenosis rate 95–99%, nearly occluded).

All participants underwent carotid vessel wall MRI on a 3.0-T MR scanner (uMR780, United Imaging Healthcare, Shanghai, China) with an 8-channel dedicated carotid coil. The specific magnetic resonance imaging scheme has been described in previous studies [15].

Carotid plaque traits (including morphological features and compositional features) represented by the images in the four sequences above were interpreted using Vascular Explorer 2 software (TSimaging Healthcare, Beijing, China). The exact method of manually outlining the boundaries of the lumen, wall, and plaque components at each axial MR image of the carotid artery has been described in previous studies [15].

### 2.3. Immunohistochemistry

A total of 20 cases of carotid plaques were included which were obtained by randomly selecting 4 cases from each of the 5 groups. This study assessed the extent of macrophage expression by measuring CD68 expression [16].

Carotid plaques were collected immediately after CEA, fixed in 4% paraformaldehyde, and embedded in paraffin for immunohistochemical staining; specific immunohistochemical steps can be found in the Appendix A.

All tissue sections were observed under a THUNDER imager DMI8 (Leica, Wetzlar, Germany), and all tissue sections were observed at the same magnification (10×) and using the same observation parameters.

Average optical density (AOD) used to represent the protein expression per area within the observed section. The AODs of 4 different proteins (H-FT, L-FT, TfR1, and CD68) were calculated for 20 sections using ImageJ software (1.53q; NIH, Bethesda, MD, USA).

### 2.4. Statistical Analysis

Normally distributed continuous variables are expressed as the means and standard deviations, and nonnormally distributed variables are expressed as medians and interquartile ranges. Categorical variables are described as counts and percentages. The independent *t*-test, Mann‒Whitney U test, chi-square test, and one-way analysis of variance approaches were used to identify any significant differences between baseline clinical characteristics in the 5 groups and any differences in AOD of the 4 different deposited proteins in the 5 groups. Ordered multiclass logistic regression analysis was used to analyze the correlations between the graded degree of carotid plaque stenosis rate and serum iron metabolism index. Correlation analysis was used to investigate the correlation between serum iron metabolism indicators and carotid plaque traits in patients, the expression of 4 different proteins and their correlation with carotid plaque traits, and correlations between serological iron metrics and the expression of the 4 different proteins. Multiple linear regression analysis was used to study the relationship between NWI and serum iron metabolism index and basic characteristics. Statistical analysis was performed using IBM SPSS Statistics 26.0 (SPSS Inc., Chicago, IL, USA). Statistical significance was considered when *p* < 0.05 (two-tailed).

## 3. Results

### 3.1. Clinical Characteristics of the Study Population

A total of 100 patients (mean age: 65.9 ± 7.8 years; 83 males) were included and had a 100% surgical success rate with no adverse events such as cerebrovascular accidents, neurological dysfunction, or death. Patients were divided into 5 groups according to the degree of stenosis: 15 in group A, 30 in group B, 33 in group C, 13 in group D, and 9 in group E. There was a statistically significant difference in the UIBC (*p* = 0.006) and CK-MB (*p* = 0.021) among patients with different carotid stenosis grades. The remaining differences in clinical characteristics among the different groups were not statistically significant, as shown in Table 1.

### 3.2. Ordered Multiclass Logistic Regression Analysis of Stenosis Grading and Serum Iron Metabolism Indexes

The results showed that no significant correlation between high SF, UIBC and progression of carotid stenosis (OR 1.100, 95% CI 0.004–0.165, *p* = 0.039; OR 1.050, 95% CI 0.005–0.094, *p* = 0.031), as shown in Table 2.

### 3.3. Correlation Analysis between the Serum Iron Metabolism Index and Carotid Plaque Traits

Nineteen of twenty plaques randomly sampled in the subgroup were included in the study. One case was discarded because the tissue had many calcified components, and very little tissue was left for immunohistochemical analysis after decalcification. The morphological characteristics of the carotid plaques and the distribution of the different components within the plaques are presented in Figure 1.

Table 3 summarizes the results of the linear regression analysis between serum iron metabolism indicators and carotid plaque characteristics revealed by hr-MRI. Among these, correlations were found between SF and arterial lumen volume (R = 0.522, *p* = 0.018) and NWI (R = 0.470, *p* = 0.036). sTfR was correlated with arterial vessel wall volume (R = 0.521, *p* = 0.018), arterial vessel wall area (R = 0.481, *p* = 0.032), arterial wall thickness (R = 0.488, *p* = 0.030) and the NWI (R = 0.449, *p* = 0.046).

Table 4 summarizes the multiple linear regression analysis of NWI with serum iron metabolism index and basic characteristics. SF and sTfR were significantly associated with NWI. In further analyses, SF and sTfR were still significantly associated with NWI in multiple linear regression analyses, and the regression models were statistically significant (R = 0.63, *p* = 0.014).

### 3.4. Analysis of Differential Protein Expression in 20 Cases of Carotid Plaques

The expression results of four different proteins in carotid plaques are shown in Figure 2A (*n* = 5). The results of the analysis of different protein expression levels in 20 cases of carotid plaques are summarized in Table 5 and Figure 2B. The results showed that the differences in the expression of four proteins, H-FT, L-FT, TfR1, and CD68, were statistically significant among carotid plaques with different degrees of stenosis (*p* < 0.001; *p* = 0.001; *p* = 0.032; *p* = 0.005). Among the three proteins associated with iron metabolism, their expression increased with increasing stenosis, with H-FT having a higher expression than L-FT and TfR1. In addition, no correlations were found between the expression levels of H-FT, L-FT, and TfR1 and their corresponding serological indexes.

Table 6 summarizes the results of the linear regression analysis between the expression of iron metabolizing proteins in plaques and the carotid plaque characteristics revealed by hr-MRI. In the analysis of the volume loading characterization of the plaques, the results showed that the expression of H-FT and CD68 was positively correlated with NWI (R = 0.502, *p* = 0.028; R = 0.590, *p* = 0.008). The expression of L-HF and TfR1 did not correlate with NWI. In the analysis of the component characterization of the plaques, the results showed that the expression of H-FT, L-FT and TfR1 was correlated with LRNC volume (R = 0.468, *p* = 0.043; R = 0.546, *p* = 0.016; R = 0.496, *p* = 0.031); additionally, there was a correlation between the expression of H-FT, L-FT, TfR1 and IPH volume (R = 0.553, *p* = 0.014; R = 0.570, *p* = 0.011; R = 0.642, *p* = 0.003).

## 4. Discussion

Iron metabolism is closely linked to the progression of atherosclerosis. When iron is overloaded, it can react with cells to produce hydroxyl radicals which, in turn, promote ferroptosis and accelerate the accumulation of lipid oxides [17,18]. In contrast, ferritin can be regulated by iron overload and oxidative stress, and its main function is to respond to iron overload and reactive oxygen species [19]. In this study, we investigated the differences in serum iron metabolism indexes, as well as the types and differences in the expression of iron metabolizing proteins in plaques in patients with different degrees of stenosis. We also investigated the correlation between serum and plaque iron metabolism indicators, and plaque vulnerability indicators.

There is no conclusive evidence as to whether serum iron metabolism indicators can accurately predict the progression of atherosclerosis-related diseases. A prospective study illustrates that elevated SF are associated with 5-year progression of carotid atherosclerosis [2]. However, other studies have concluded that SF, transferrin, or dietary iron levels are not associated with carotid atherosclerosis progression [20]. In the present study, the results of ordered multiclass logistic regression analysis of stenosis grading and serum iron metabolic indexes in Table 2 showed that the ORs for SF, UIBC converged to 1 (1.100 and 1.050), and we concluded that high SF and high UIBC levels were not associated with progression of carotid stenosis to occlusion; however, as shown in Table 4, the results of multiple linear regression analysis of NWI with serum iron metabolism indexes and basic characteristics showed significant correlations between SF, sTfR, and NWI, suggesting that SF and sTfR may be serologic predictors of plaque vulnerability. In addition, inflammation and oxidative stress are recognized as important factors in atherosclerosis, but this study did not find differences in serum inflammatory indices (CRP) among patients with different degrees of stenosis. It has been shown that higher levels of inflammatory biomarker values before CEA (systemic inflammation index, systemic inflammatory response index, systemic inflammation synthesis index, etc.) can predict restenosis rate and mortality within 12 months after surgery [21]; and that higher plasma malondialdehyde levels, which are associated with the risk of stroke, are a reliable marker of carotid plaque vulnerability [22]. This provides us with ideas for future studies to understand whether preoperative inflammatory biomarker levels and oxidative stress marker levels are associated with carotid plaque vulnerability.

In the present study, there were statistical differences in the expression of iron metabolism proteins in carotid plaques with different degrees of stenosis. We found that H-FT, L-FT, and TfR1 were all expressed in carotid plaques, and their expression increased with increasing stenosis. In addition, the expression of H-FT was higher than that of L-FT and TfR1. Thus, as the degree of carotid stenosis increased, the expression of H-FT, L-FT, and TfR1 in plaques increased, but the corresponding serum iron metabolism indexes did not have the same incremental trend, which suggests that, with the progression of carotid atherosclerotic disease, the changes in serum iron metabolism indexes may not be the same as those in the expression of iron metabolizing proteins in plaques. This result may provide some ideas for subsequent mechanistic studies.

Carotid plaque vulnerability is determined by both the volume loading characteristics of the carotid plaque and the carotid plaque composition characteristics. MRI, with its high soft-tissue contrast and high in-plane resolution, has a good detection effect for morphometric measurements of carotid vessels and identification of the components of vulnerable plaques, so the use of hr-MRI to quantify the vulnerability of carotid plaques in this study was accurate and reliable [23]. We found that changes in iron metabolism in serum and in plaques were positively correlated with indicators of plaque vulnerability (NWI, LRNC volume). First, the present study found that SF and sTfR in serum were positively correlated with NWI (structural features of vulnerable plaques). It has been noted in the literature that the severity of NWI significantly correlates with the volume loading of carotid plaque and is a better indicator of the severity of atherosclerotic disease than the degree of stenosis [24,25]. Second, the expression of H-FT, L-FT, and TfR1 in plaque was positively correlated with LRNC volume, a component characteristic of vulnerable plaques. LRNC is an important factor leading to plaque rupture to form vulnerable plaques and may also reflect the volume load of plaques to some extent [6]. This shows that there is a correlation between iron metabolism-related indices and carotid plaque vulnerability indices. It is noteworthy that the expression of H-FT in plaques was positively correlated with both NWI and LRCN volume, which represents H-FT as an important iron metabolizing protein that can characterize carotid plaque vulnerability. This is consistent with our previous animal studies [26], in which when the expression of H-FT was reduced in the arteries of APOE knockout mice, their NWI and the expression of metalloproteinases, which represents arterial plaque vulnerability, were reduced.

The present study had several limitations. First, the sample size of the included patients and the sample size used for quantitative analysis of carotid plaque traits and iron metabolism protein expression was small, and the sample size will be further expanded to improve studies in the future. Second, to completely remove the carotid plaque and the thickened carotid intima, the surgeon was forced to cut the arterial wall along the long axis of the carotid artery during the procedure, which resulted in not obtaining complete long tubular plaque tissue; moreover, this did not ensure the morphological similarity of all samples during immunohistochemical sectioning.

## 5. Conclusions

There were statistical differences in the expression of iron metabolism proteins in carotid plaques with different degrees of stenosis. The serum iron metabolism index (SF and sTfR) and expression of iron metabolizing proteins (H-FT and TfR1) in plaques were positively correlated with carotid plaque vulnerability index (NWI, LRNC volume).

## Figures and Tables

**Figure 1 diagnostics-13-03196-f001:**
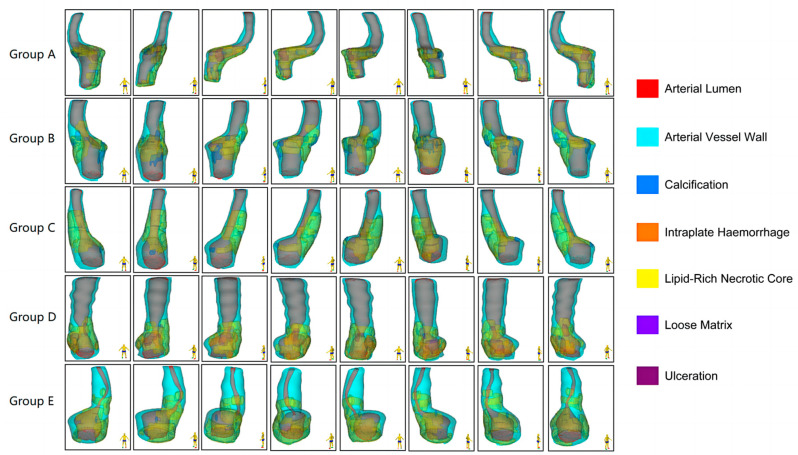
Multidimensional quantitative illustration of the intraplaque components of carotid plaques with different degrees of stenosis (*n* = 5). Demonstrating the morphological characteristics of the plaques and the distribution of the different components within the plaques from eight directions.

**Figure 2 diagnostics-13-03196-f002:**
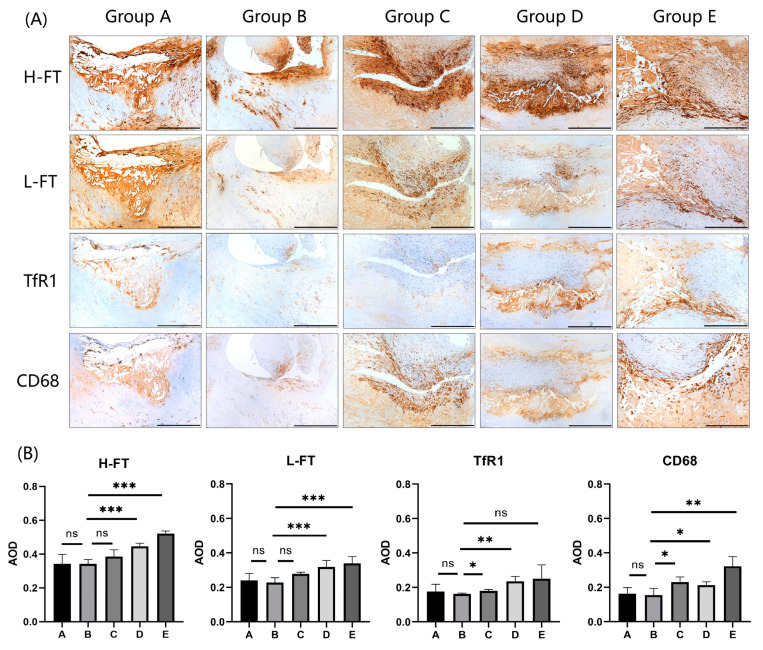
Expression analysis of H-FT, L-FT, and TfR1 in carotid plaques with different degrees of stenosis. (**A**) Immunohistochemical results showing that the iron metabolism proteins H-FT, L-FT, and TfR1, which are consistent with CD68 expression sites in carotid plaques, are mainly distributed in the carotid intima and carotid mesentery (*n* = 5). Positive staining was brownish yellow. Bar = 500 μm. (**B**) Graphs showing the results of immunohistochemical AOD value analysis of H-FT, L-FT, TfR1, and CD68 in carotid plaques in a total of 20 cases from 5 groups. * *p* < 0.05, ** *p* < 0.01, *** *p* < 0.001, ns indicates no significant differences.

**Table 1 diagnostics-13-03196-t001:** Clinical characteristics of the study population (*n* = 100).

Indexes	Mean ± SD, *n* (%)						*p*-Value
	All Patients (*n* = 100)	Group A (*n* = 15)	Group B (*n* = 30)	Group C (*n* = 33)	Group D (*n* = 13)	Group E (*n* = 9)	
**Age**, years	65.9 ± 7.8	67.9 ± 1.3	65.5 ± 1.6	66.2 ± 1.3	63.5 ± 2.0	66.6 ± 0.8	0.672
**SF**, µmol/L	18.6 ± 7.5	22.0 ± 10.0	18.8 ± 6.4	16.5 ± 6.6	17.8 ± 7.1	21.3 ± 8.0	0.127
**sTfR**, nmol/L	38.3 ± 28.4	35.5 ± 9.2	33.0 ± 11.8	43.2 ± 43.9	45.5 ± 27.7	32.6 ± 13.0	0.505
**UIBC**, µmol/L	37.3 ± 11.4	31.5 ± 10.5	36.2 ± 8.9	41.1 ± 11.7	42.1 ± 14.3	30.0 ± 6.0	0.006
**cTnT**, ng/mL	0.02 ± 0.05	0.04 ± 0.10	0.01 ± 0.01	0.02 ± 0.05	0.02 ± 0.02	0.02 ± 0.03	0.330
**CRP**, mg/L	6.3 ± 16.5	4.7 ± 9.3	6.1 ± 9.0	9.1 ± 26.5	3.6 ± 4.4	3.3 ± 4.4	0.792
**CK-MB**, U/L	48.3 ± 44.5	54.4 ± 37.6	35.3 ± 26.0	55.3 ± 57.2	29.6 ± 18.4	82.3 ± 44.5	0.021
**Serum Cholesterol**, mmol/L	4.1 ± 1.1	3.9 ± 1.0	4.0 ± 1.0	4.4 ± 1.2	4.3 ± 1.0	3.8 ± 1.2	0.363
**Serum Triglyceride**, mmol/L	1.4 ± 0.7	1.4 ± 0.7	1.3 ± 0.6	1.5 ± 0.8	1.4 ± 0.7	1.6 ± 0.6	0.787
**Glycosylated Hemoglobin**, g/dL	6.7 ± 1.1	6.8 ± 1.4	6.5 ± 0.8	6.7 ± 1.2	6.5 ± 0.8	6.9 ± 1.1	0.853
**Homocysteine**, μmol/L	15.5 ± 8.4	14.6 ± 8.5	15.8 ± 8.5	17.0 ± 9.5	12.9 ± 4.8	14.6 ± 8.8	0.660
**Gender**, male	83 (83.0)	13 (86.7)	25 (83.3)	25 (75.8)	11 (84.6)	9 (100)	0.326
**Hypertension**, no	36 (36.0)	3 (20.0)	11 (36.7)	12 (36.4)	5 (38.5)	5 (55.6)	0.510
**Diabetes**, no	55 (55.0)	8 (53.3)	18 (60.0)	19 (57.6)	5 (38.5)	5 (55.6)	0.764
**Coronary Disease**, no	84 (84.0)	11 (73.3)	24 (80.0)	29 (87.9)	12 (92.3)	8 (88.9)	0.591
**Hyperlipidemia**, no	65 (65.0)	11 (73.3)	19 (63.3)	22 (66.7)	8 (61.5)	5 (55.6)	0.915
**Hyperhomocysteinemia**, no	67 (67.0)	10 (66.7)	21 (70.0)	20 (60.6)	9 (69.2)	7 (77.8)	0.869
**History of Stroke**, no	72 (72.0)	12 (80.0)	18 (60.0)	27 (81.8)	9 (69.2)	7 (77.8)	0.497
**History of Smoke**, no	82 (82.0)	12 (80.0)	25 (83.3)	27 (81.8)	12 (92.3)	6 (66.7)	0.657
**History of Alcohol**, no	93 (93.0)	14 (93.3)	30 (100.0)	29 (87.9)	13 (100)	7 (77.8)	0.100
**Blood Type**							0.608
A-Type	25 (25.0)	5 (33.3)	11 (36.7)	7 (21.2)	1 (7.7)	1 (11.1)	
B-Type	29 (29.0)	4 (26.7)	6 (20.0)	11 (33.3)	6 (46.2)	2 (22.2)	
AB-Type	16 (16.0)	3 (20.0)	4 (13.3)	4 (12.1)	2 (15.4)	3 (33.3)	

SF = serum ferritin; sTfR = serum transferrin receptor; UIBC = unsaturated iron-binding capacity; cTnT = cardiac troponin T; CRP = C-reactive protein; CK-MB = creatine kinase isoenzyme MB.

**Table 2 diagnostics-13-03196-t002:** Ordered multiclass logistic regression analysis of stenosis grading and serum iron metabolic indexes.

Indexes	*p*-Value	OR	95% CI
**Age**, years	0.507	1019	−0.038–0.076
**SF**, µmol/L	0.039	1.100	0.004–0.165
**sTfR**, nmol/L	0.689	1.004	−0.014–0.022
**UIBC**, µmol/L	0.031	1.050	0.005–0.094
**cTnT**, ng/mL	0.237	0.050	−14.104–3.494
**CRP**, mg/L	0.843	1.003	−0.028–0.034
**CK-MB**, U/L	0.171	1.008	−0.004–0.021
**Serum Cholesterol**, mmol/L	0.499	0.850	−0.633–0.308
**Serum Triglyceride**, mmol/L	0.483	1.293	−0.461–0.974
**Glycosylated Hemoglobin**, g/dL	0.255	1.339	−0.796–0.212
**Homocysteine**, μmol/L	0.357	0.970	−0.093–0.033
**Gender**, male	0.625	1.362	−0.931–1.549
**Hypertension**, no	0.503	1.324	−0.542–1.104
**Diabetes**, no	0.322	0.591	−1.567–0.516
**Coronary Disease**, no	0.295	1.925	−0.570–1.881
**Hyperlipidemia**, no	0.786	1.153	−0.885–1.169
**Hyperhomocysteinemia**, no	0.820	1.477	−1.065–0.344
**History of Stroke**, no	0.589	0.260	−6.226–3.535
**History of Smoke**, no	0.158	2.940	−0.362–2.230
**History of Alcohol**, no	0.018	0.093	−4.341–−0.400

SF = serum ferritin; sTfR = serum transferrin receptor; UIBC = unsaturated iron-binding capacity; cTnT = cardiac troponin T; CRP = C-reactive protein; CK-MB = creatine kinase isoenzyme MB.

**Table 3 diagnostics-13-03196-t003:** The linear regression analysis between serum iron metabolism indexes and carotid plaque traits.

Indexes	SF		sTfR		TIBC		UIBC	
	R	*p*-Value	R	*p*-Value	R	*p*-Value	R	*p*-Value
**Arterial lumen volume**, mm^3^	0.522	0.018	0.026	0.914	0.008	0.973	0.439	0.053
**Arterial lumen area**, mm^2^	0.272	0.246	0.110	0.643	0.066	0.781	0.201	0.396
**Arterial vessel wall volume**, mm^3^	0.118	0.622	0.521	0.018	0.127	0.594	0.169	0.476
**Arterial vessel wall area**, mm^2^	0.198	0.402	0.481	0.032	0.027	0.910	0.148	0.534
**Arterial volume**, mm^3^	0.409	0.074	0.358	0.122	0.078	0.744	0.390	0.090
**Arterial area**, mm^2^	0.009	0.972	0.381	0.098	0.016	0.948	0.006	0.980
**Arterial wall thickness**, mm	0.427	0.060	0.488	0.030	0.234	0.320	0.231	0.328
**NWI**, %	0.470	0.036	0.449	0.047	0.176	0.458	0.299	0.201
**LRNC maximum area percentage**, %	0.213	0.368	0.033	0.888	0.020	0.934	0.186	0.432
**LRNC volume**, mm^3^	0.038	0.874	0.250	0.288	0.076	0.748	0.091	0.704
**Ulcer maximum area percentage**, %	0.007	0.978	0.227	0.334	0.431	0.058	0.172	0.468
**Ulcer volume**, mm^3^	0.095	0.690	0.311	0.182	0.367	0.112	0.186	0.432
**IPH maximum area percentage**, %	0.107	0.652	0.003	0.988	0.058	0.808	0.048	0.840
**IPH volume**, mm^3^	0.110	0.644	0.038	0.874	0.083	0.730	0.042	0.860
**Calcification maximum area percentage**, %	0.119	0.618	0.041	0.866	0.187	0.430	0.191	0.420
**Calcification volume**, mm^3^	0.219	0.354	0.052	0.828	0.215	0.364	0.292	0.212

SF = serum ferritin; sTfR = serum transferrin receptor; UIBC = unsaturated iron-binding capacity; TIBC = total iron binding capacity; NWI = normalized wall index; LRNC = lipid-rich necrotic core; IPH = intraplate hemorrhage.

**Table 4 diagnostics-13-03196-t004:** Multiple linear regression analysis of NWI with serum iron metabolism index and basic characteristics.

Indexes	Univariate Regression			Multivariate Regression Model *		
	R	Beta (95% CI)	*p*-Value	Beta (95% CI)	*p*-Value	Standardized Beta
**Age**, years	0.196	0.216 (−0.320–0.751)	0.409	-	-	-
**Gender**, male	0.089	3.558 (−16.253–23.369)	0.701	-	-	-
**SF**, µmol/L	0.470	−0.460 (−0.888–−0.033)	0.036	−0.433 (−0.823–−0.044)	0.031	−0.443
**sTfR**, nmol/L	0.449	0.418 (0.006–0.829)	0.047	0.391 (0.021–0.761)	0.040	0.420
**UIBC**, µmol/L	0.299	0.261 (−0.152–0.673)	0.201	-	-	-
**TIBC**, µmol/L	0.176	−0.285 (−1.075–0.504)	0.458	-	-	-

SF = serum ferritin; NWI = normalized wall index; sTfR = serum transferrin receptor; UIBC = unsaturated iron-binding capacity; TIBC = total iron binding capacity. * Inclusion of variables: SF, sTfR. Appendix A shows a summary of the multiple linear regression model results (R = 0.630, R^2^ = 0.397, adjusted R^2^ = 0.326, *p* = 0.014).

**Table 5 diagnostics-13-03196-t005:** Analysis of different protein expression in 20 cases of carotid plaque.

Indexes	All Patients’ AOD (*n* = 19)	Group A (*n* = 3)	Group B (*n* = 4)	Group C (*n* = 4)	Group D (*n* = 4)	Group E (*n* = 4)	*p*-Value
**H-FT**	0.411 ± 0.076	0.343 ± 0.055	0.342 ± 0.026	0.385 ± 0.040	0.445 ± 0.191	0.523 ± 0.150	<0.001
**L-FT**	0.283 ± 0.052	0.240 ± 0.040	0.227 ± 0.028	0.278 ± 0.009	0.318 ± 0.039	0.340 ± 0.038	0.001
**TfR1**	0.202 ± 0.053	0.177 ± 0.042	0.162 ± 0.005	0.180 ± 0.008	0.235 ± 0.029	0.250 ± 0.081	0.032
**CD68**	0.219 ± 0.070	0.163 ± 0.035	0.155 ± 0.038	0.230 ± 0.029	0.213 ± 0.190	0.323 ± 0.056	0.028

H-FT = ferritin H; L-FT = ferritin L; TfR1 = transferrin receptor 1; AOD = average optical density; CD 68 = Cluster of Differentiation 68.

**Table 6 diagnostics-13-03196-t006:** Linear regression analysis between AOD of iron metabolizing proteins and carotid plaque characteristics.

Indexes	H-FT		L-FT		TfR1		CD-68	
	R	*p*-Value	R	*p*-Value	R	*p*-Value	R	*p*-Value
**Arterial lumen volume**, mm^3^	0.274	0.075	0.203	0.405	0.255	0.292	0.466	0.044
**Arterial lumen area**, mm^2^	0.085	0.729	0.060	0.807	0.004	0.988	0.299	0.213
**Arterial vessel wall volume**, mm^3^	0.256	0.289	0.168	0.493	0.001	0.996	0.314	0.191
**Arterial vessel wall area**, mm^2^	0.359	0.131	0.359	0.131	0.227	0.349	0.406	0.084
**Arterial volume**, mm^3^	0.005	0.984	0.019	0.937	0.179	0.464	0.097	0.693
**Arterial area**, mm^2^	0.219	0.368	0.296	0.218	0.169	0.488	0.140	0.567
**Arterial wall thickness**, mm	0.315	0.189	0.189	0.439	0.144	0.557	0.070	0.185
**NWI**, %	0.502	0.028	0.327	0.172	0.199	0.414	0.590	0.008
**LRNC maximum area percentage**, %	0.437	0.062	0.439	0.060	0.479	0.038	0.549	0.015
**LRNC volume**, mm^3^	0.468	0.043	0.546	0.016	0.496	0.031	0.638	0.003
**Ulcer maximum area percentage**, %	0.097	0.693	0.014	0.954	0.162	0.678	0.157	0.520
**Ulcer volume**, mm^3^	0.057	0.817	0.051	0.835	0.093	0.705	0.177	0.468
**IPH maximum area percentage**, %	0.411	0.081	0.447	0.055	0.625	0.002	0.629	0.004
**IPH volume**, mm^3^	0.553	0.014	0.570	0.011	0.642	0.003	0.736	<0.01
**Calcification maximum area percentage**, %	0.153	0.532	0.126	0.608	0.216	0.375	0.025	0.919
**Calcification volume**, mm^3^	0.140	0.567	0.102	0.677	0.213	0.381	0.049	0.842

H-FT = ferritin H; L-FT = ferritin L; TfR1 = transferrin receptor 1; AOD = average optical density; LRNC = lipid-rich necrotic core; IPH = intraplate hemorrhage.

## Data Availability

All data generated or analyzed during this study are included in this article. Further inquiries can be directed to the corresponding author.

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
