# Peer review of "The Relation of the Iron Metabolism Index to the Vulnerability Index of Carotid Plaque with Different Degrees of Stenosis"

_diagnostics, 2023, doi:10.3390/diagnostics13203196_

Round 1

Reviewer 1 Report

Poor quality of illustrations

Author Response

Reviewer #1:

Reviewer #1, Comment #1:

Poor quality of illustrations.

Author response: 

Thank you for the comments.

For Figure 1, we apologize for the lack of clarity of the carotid plaque feature images produced by the vessel Explorer 2 software (TSimaging Healthcare, Beijing, China). However, this is the highest resolution image that the software can produce for carotid plaque reconstruction (showing the morphological and compositional characteristics of carotid plaques from 8 directions). We have tried our best to adjust for the higher resolution, but the blurring of the image due to color overlay is still unavoidable. Again, we apologize for this, but we have re-uploaded the highest resolution version of the image we were able to resize. We have updated a higher resolution version of Figure 1.

For Figure 2, we have updated a higher resolution version.

Author action:

We have updated a higher resolution version of Figure 1 and Figure 2. on Page 7, Lines 171 and Page 10, Lines 209.

--------------------------------------------------

Reviewer 2 Report

The manuscript described association of iron metabolism indexes and iron metabolizing proteins with carotid artery stenosis and their correlation with plaque vulnerability indexes using clinical samples. The authors have done a difficult job, but the presentation of the data and their discussion raise questions - and there are doubts about the adequacy of the results. A close reading of the materials and methods, and results sections raised the following questions:

1.     The control arm is missing and raises a big question on the validity of this study. The control arm should be included to better understand the results.

2.     The authors have over-claimed the results, the odds ratio for SF and UIBC are 1.1 and 1.05 which are just above 1, which is not a high odds ratio. And again, in absence of appropriate control, it doesn’t make sense. Group A and E have high SF while Group B, C and D have lower than average SF value, so, how is it associated with progression, it seems very random as is defined by p value (0.127).
For UIBC, the increasing trend from Group A to D is there while there is sudden decrease in group E, in fact, Group E has lowest values for UIBC.

3.     The authors have measured the SF, sTFR, UIBC and TIBC from serum but the methodology is not explained. How did they measure these indexes?

4.     In the abstract, it is mentioned that “H-FT was correlated with arterial volume and NWI and LRNC volume (r=0.502, p= 0.028; r= 0.468, p= 0.043)”. First of all, there are only two values for three correlations and more importantly, where are these results? These data are not available in any table.

5.     Similarly, “TfR1 was correlated with LRNC volume and IPH volume (r= 0.496, p= 0.031; r= 0.642, p= 0.003)”. Where is this data, it is not available in manuscript?

6.     The same results are mentioned in results section line 206 to 214.  Where is the data for these results?

7.     The carotid plaques were used for immunohistochemistry, why are these plaque sections were incubated with serum, please explain.

8.     Again the statement in line 237 “Thus, with the increase……..as well as the corresponding serum iron metabolism indexes (SF, UIBC) in the plaques also increased” does not corroborate with data shown in table 1. (mentioned in point 2 also).

9.     As authors have rightly mentioned, the sample size for carotid plaques’ study was small and if possible, they should include more samples.

The English is fine, minor improvements are needed.

Author Response

Reviewer #2:

Reviewer #2, Comment #1:

The control arm is missing and raises a big question on the validity of this study. The control arm should be included to better understand the results.

Author response: 

Thank you for your comments. The control group setting is important. However, for this study, it was difficult to select control patients because we could not obtain carotid plaques with low stenosis that did not meet the surgical indications in the clinic, and non-vulnerable carotid plaques rarely caused clinical symptoms. Therefore, in this study, we categorized patients according to the degree of carotid stenosis, and there was a sequential relationship between the categories.

In the analysis of the relationship between serum iron metabolism indices and carotid stenosis grading, we used Ordered Multiclass Logistic Regression Analysis to explore the serological iron metabolism indices associated with the progression of carotid stenosis grading. The principle of Ordered Multiclass Logistic Regression Analysis is to split multiple classifications of the dependent variable sequentially into binary logistic regression analysis. In the present study, this statistical method is to compare the low stenosis group (Groups A, B, C, and D) with the high stenosis group (Group E) sequentially and output the coefficients of the independent variables for one group, which ensures the validity of the research results.

Reviewer #2, Comment #2:

The authors have over-claimed the results, the odds ratio for SF and UIBC are 1.1 and 1.05 which are just above 1, which is not a high odds ratio. And again, in absence of appropriate control, it doesn’t make sense. Group A and E have high SF while Group B, C and D have lower than average SF value, so, how is it associated with progression, it seems very random as is defined by p value (0.127).

For UIBC, the increasing trend from Group A to D is there while there is sudden decrease in group E, in fact, Group E has lowest values for UIBC.

Author response: 

Thank you for your comment. We have reanalyzed the results of Table 2, and the statement about ‘High serum ferritin (SF) and unsaturated iron-binding capacity (UIBC) were associated with the progression of carotid plaque.’ is inappropriate.

We have updated the relevant content in the ‘Abstract’ ‘Results’‘Discussion’and ‘Conclusion’ sections.

Author action:

We have updated the content as ‘No significant correlation between high serum ferritin (SF), unsaturated iron binding capacity (UIBC) and progression of carotid stenosis’ (Page 1, Lines 16-17), and ‘There were statistical differences in the expression of iron metabolism proteins in carotid plaques with different degrees of stenosis. Serum iron metabolism index (SF and sTfR) and expression of iron metabolizing proteins (H-FT and TfR1) in plaques were positively correlated with carotid plaque vulnerability index (NWI, LRNC volume).’ (Page 1, Lines 26-29) in the ‘Abstract’ section.

We have updated the content as ‘The results showed that no significant correlation between high SF, UIBC and progression of carotid stenosis’ (Page 5, Lines 155-156) in the ‘’Results’ section.

We have updated the content as ‘In the present study, the results of ordered multiclass logistic regression analysis of stenosis grading and serum iron metabolic indexes in Table 2 showed that the ORs for SF, UIBC converged to 1, (1.100 and 1.050), and we concluded that high SF and high UIBC levels were not associated with progression of carotid stenosis to occlusion; however, as shown in Table 4, the results of multiple linear regression analysis of NWI with serum iron metabolism indexes and basic characteristics showed significant correlations between SF, sTfR, and NWI, suggesting that SF and sTfR may be serologic predictors of plaque vulnerability.’ (Page 12, Lines 251-258) in the ‘Discussion’ section.

We have updated the content as ‘There were statistical differences in the expression of iron metabolism proteins in carotid plaques with different degrees of stenosis. Serum iron metabolism index (SF and sTfR) and expression of iron metabolizing proteins (H-FT and TfR1) in plaques were positively correlated with carotid plaque vulnerability index (NWI, LRNC volume).’ (Page 13, Lines 318-321) in the ‘Conclusion’ section.

Reviewer #2, Comment #3:

The authors have measured the SF, sTFR, UIBC and TIBC from serum but the methodology is not explained. How did they measure these indexes?

Author action: 

Thank you for your comments.

We have updated the content as ‘The included patients underwent routine blood tests before CEA (including blood routine, blood biochemistry, iron metabolism indicators, etc.), and the samples will be sent to the clinical laboratory of the hospital for standardized testing. In this study, the ranges of the four serum iron metabolism indices were as follows: SF 10.6-36.7µmol/L; sTfR 20.8-59.16 nmol/L; total iron binding capacity (TIBC) 46.4-69.6µmol/L; unsaturated iron-binding capacity (UIBC) 31-48µmol/L.’ on Page 2, Lines 73-77.

Reviewer #2, Comment #4:

In the abstract, it is mentioned that “H-FT was correlated with arterial volume and NWI and LRNC volume (r=0.502, p= 0.028; r= 0.468, p= 0.043)”. First of all, there are only two values for three correlations and more importantly, where are these results? These data are not available in any table.

Reviewer #2, Comment #5:

Similarly, “TfR1 was correlated with LRNC volume and IPH volume (r= 0.496, p= 0.031; r= 0.642, p= 0.003)”. Where is this data, it is not available in manuscript?

Reviewer #2, Comment #6:

The same results are mentioned in results section line 206 to 214.  Where is the data for these results?

Author response: 

Thank you for your comments. We apologize for the unclear description of the results and the lack of relevant tables. We have modified the relevant content in the ‘Abstract’ and added Table 6.

Author action:

We have updated the content as ‘SF and serum transferrin receptor (sTfR) were correlated with normalized wall index (NWI) (R = 0.470, p = 0.036; R = 0.449, p = 0.046), and the results of multiple linear regression suggested that SF and sTfR remained associated with NWI ((R = 0.630, R2 = 0.397, Adjusted R2 = 0.326, p = 0.014).’ (Page 1, Lines 21-23) in the ‘Abstract’ section.

We have updated the content as ‘Table 6 summarizes the results of the linear regression analysis between the expression of iron metabolizing proteins in plaques and the carotid plaque characteristics revealed by hrMRI.’ in Page 11, Lines 220-222, added Table 6 in Page 11, Lines 231-235.

Reviewer #2, Comment #7:

The carotid plaques were used for immunohistochemistry, why are these plaque sections were incubated with serum, please explain.

Author action:

Thank you for your comments. We apologize for the unclear description of the immunohistochemistry.

We have changed ‘After incubating the sections with serum, the sections were exposed to the following primary antibodies: monoclonal rabbit anti-ferritin heavy chain’ to ‘Sections were blocked with serum and then the sections were exposed to the following primary antibodies: monoclonal rabbit anti-ferritin heavy chain’ in the 'Supplementary Material'.

Reviewer #2, Comment #8:

Again the statement in line 237 “Thus, with the increase……..as well as the corresponding serum iron metabolism indexes (SF, UIBC) in the plaques also increased” does not corroborate with data shown in table 1. (mentioned in point 2 also).

Author response: 

Thank you for your comments. Your useful review has helped us avoid mischaracterizing the results. We have modified the relevant content in the "Discussion" section.

Author action:

We have updated the content as ‘Thus, as the degree of carotid stenosis increased, the expression of H-FT, L-FT, and TfR1 in plaques increased, but the corresponding serum iron metabolism indexes did not have the same incremental trend, which suggests that, with the progression of carotid atherosclerotic disease, the changes in serum iron metabolism indexes may not be the same as those in the expression of iron metabolizing proteins in plaques.’ (Page 12, Lines 276-280) in the 'Discussion' section.

Reviewer #2, Comment #9:

As authors have rightly mentioned, the sample size for carotid plaques’ study was small and if possible, they should include more samples.

Author response: 

Thank you for your comments. The quantitative imaging analysis of carotid plaque traits and the quantitative analysis of iron metabolizing protein expression used in this study require a high economic cost, which not allow us to achieve larger sample sizes at this time. The sample size will be further expanded to improve studies in the future. The small sample size of patients enrolled in this study has been as a limitation and specified in the "Discussion" section. We have updated the content as ‘First, the sample size of the included patients and the sample size used for quantitative analysis of carotid plaque traits and iron metabolism protein expression was small, and the sample size will be further expanded to improve studies in the future.’ in Page 13, Lines 309-310.

--------------------------------------------------

Reviewer 3 Report

Dear Authors,

I propose your paper to be accepted in present form,

but suppose that References need to be continued

(e.g.23 authors are not completed?).

Your contributions on iron-metabolism related indices and carotid artery plaques vulnerable indices could improve surgical therapeutic results (subsequent mechanistics),  and especially -  patients-individual treatment/management of disease.

Additional question also arrised from your original immunocytochemical studies: correlation between reduced macrophageal iron methabolism and carotid plaques vulnerably indices? Best regards! Reviewer

Author Response

Reviewer #3:

Reviewer #3, Comment #1:

I propose your paper to be accepted in present form, but suppose that References need to be continued (e. g. 23 authors are not completed?).

Your contributions on iron-metabolism related indices and carotid artery plaques vulnerable indices could improve surgical therapeutic results (subsequent mechanistics), and especially - patients-individual treatment/management of disease.

Additional question also arrised from your original immunocytochemical studies: correlation between reduced macrophageal iron methabolism and carotid plaques vulnerably indices? Best regards! Reviewer.

Author response: 

Thank you for your comments.

We have improved the information on the 23rd references in Page 15, Lines 404-405. Thank you for your helpful suggestions and thank you again for recognizing our research and results.

--------------------------------------------------

Reviewer 4 Report

Dear Editor,

I read with interest the article by Yuan et al. regarding the relationship between iron metabolism index and vulnerability index of carotid plaque with different degrees of stenosis. The authors' conclusion after this study was that serum iron metabolism indices and expression of iron metabolizing proteins in plaques differed among patients with different degrees of carotid stenosis. Additionally, Serum iron metabolism indices (SF and sTfR) and expression of iron metabolizing proteins in plaques were positively correlated with carotid plaque vulnerability indices (NWI, LRNC volume).

The manuscript is well written and of interest, with many variables enrolled and analyzed. However, the number of patients enrolled in this study is one of the main limitations of this study. It will be interesting if the authors can provide information of medical treatments. If not, please specify that in limitations of the study.

Also, for a study with such results, the number of references is far too small, please analyze and improve the discussion chapter by citing the following articles:

- https://doi.org/10.3390/ijerph192113934

- https://doi.org/10.3390/biom13081236

- https://doi.org/10.3390/antiox12020506

Author Response

Reviewer #4:

Reviewer #4, Comment #1:

Dear Editor,

I read with interest the article by Yuan et al. regarding the relationship between iron metabolism index and vulnerability index of carotid plaque with different degrees of stenosis. The authors' conclusion after this study was that serum iron metabolism indices and expression of iron metabolizing proteins in plaques differed among patients with different degrees of carotid stenosis. Additionally, Serum iron metabolism indices (SF and sTfR) and expression of iron metabolizing proteins in plaques were positively correlated with carotid plaque vulnerability indices (NWI, LRNC volume).

The manuscript is well written and of interest, with many variables enrolled and analyzed. However, the number of patients enrolled in this study is one of the main limitations of this study.

Author response: 

Thank you for your comments. The small sample size of patients enrolled in this study has been as a limitation and specified in the "Discussion" section.

Author action:

We have updated the content as ‘First, the sample size of the included patients and the sample size used for quantitative analysis of carotid plaque traits and iron metabolism protein expression was small, and the sample size will be further expanded to improve studies in the future.’ in Page 13, Lines 309-310.

Reviewer #4, Comment #2:

It will be interesting if the authors can provide information of medical treatments. If not, please specify that in limitations of the study.

Author response: 

We have added the medical treatments as ‘All patients took aspirin (daily doses of 100 mg, oral) before CEA and stopped taking clopidogrel 1 week before CEA. Intraoperatively, intravenous heparin (single dose of 5000µ) was routinely given before carotid artery clamping. After CEA, all patients were given heparin (2,500 IU, within 24 h), aspirin (daily dose of 100 mg, oral), and intensive statin therapy (atorvastatin, daily dose of 40 mg, oral).’ in the ‘Study sample’ of ‘Materials and Methods’ section in Page 2, Lines 85-89.

Reviewer #4, Comment #3:

Also, for a study with such results, the number of references is far too small, please analyze and improve the discussion chapter by citing the following articles:

- https://doi.org/10.3390/ijerph192113934

- https://doi.org/10.3390/biom13081236

- https://doi.org/10.3390/antiox12020506? 

Author response: 

Thanks to your comments. We have cited the above three references in the text as Reference 21, Reference 23, and Reference 22.

We have improved the ‘Discussion’ section by citing the Reference 21 in Page 12, Lines 261-267; the Reference 22 in Page 12, Lines 267-271; and the Reference 23 in Page 12, Lines 288-292.

=======================================

Round 2

Reviewer 2 Report

I want to thank the authors for incorporating the changes throughout manuscript and adding the missing data. However, still there are few points to be taken care of which are listed as below:

1. In the updated text, line 21-23, authors have mentioned “and the results of ……SF and sTFR remained associated with NWI (R=0.630……,p=0.014).” I did not find the R=0.630 in any table provided. Table 4 talks about multivariate regression and under the, there is a comment about inclusion of variables (R=0.630) but it is not shown from where this value came from, authors should mention it and show it, if not in main data, show in supplemental data.

2. Also, the univariate regression data in table 4 is repetitive of table 3.

3. The authors have added the methodology for measuring serum metabolism indices, I believe it will be better if they can mention the technique used for the measurement (by the clinical laboratory), that can be helpful for readers to understand and reproduce, if needed.

I am just curious to know why authors choose CD68 as macrophage marker among others, for this study.

Thank you.

The English seems fine to me.

Author Response

Reviewer #2:

Reviewer #2, Comment #1:

In the updated text, line 21-23, authors have mentioned “and the results of ……SF and sTFR remained associated with NWI (R=0.630……,p=0.014).” I did not find the R=0.630 in any table provided. Table 4 talks about multivariate regression and under the, there is a comment about inclusion of variables (R=0.630) but it is not shown from where this value came from, authors should mention it and show it, if not in main data, show in supplemental data.

Author response: 

Thank you for your comments.

We have summarized the essential results of the Multiple Linear Regression Model (calculated by SPSS software) and added them as Supplementary Table 1(Supplementary materials Page 3, Lines 27-30.). We have also updated the 'Results' section of the main text on Page 9, Lines 198-199.

Reviewer #2, Comment #2:

Also, the univariate regression data in table 4 is repetitive of table 3.

Author response: 

Thank you for your comments.

Table 3 shows the linear regression analysis between serum iron metabolism indexes and carotid plaque traits. NWI is included in Table 3 as one of the essential traits representing plaque vulnerability.

Table 4 shows the multiple linear regression analysis of NWI with serum iron metabolism index and basic characteristics (age and gender).

Although the results of the univariate analysis of serum iron metabolism indexes (SF, sTfR, UIBC, and TIBC) with NWI in Table 4 are duplicated in Table 3, the results of the univariate analysis of basic characteristics (age and sex) with NWI are not repeated. 

In order to provide a complete representation of the full range of indicators included in the multiple linear regression analysis model, we have listed the linear regression analysis on serum iron metabolism indexes again on the left part of Table 4 and the multiple linear regression analysis on the right part to highlight the statistically significant indicators (SF, sTfR) after multiple linear regression analysis more clearly.

Reviewer #2, Comment #3:

The authors have added the methodology for measuring serum metabolism indices, I believe it will be better if they can mention the technique used for the measurement (by the clinical laboratory), that can be helpful for readers to understand and reproduce, if needed.

Author action: 

Thank you for your comments.

We have added methods for measuring serum iron metabolism indicators in the clinical laboratory (Page 2, Lines 73-80).

Specific methods for measuring indicators of serum iron metabolism are shown in the ‘Supplementary Material Page 1, Line 1-11’.

Reviewer #2, Comment #4:

I am just curious to know why authors choose CD68 as macrophage marker among others, for this study.

Author response: 

Thank you for your comments.

CD68 is a glycosylated glycoprotein highly expressed in macrophages. Traditionally, CD68 is exploited as a valuable cytochemical marker to immunostain monocytes/macrophages in the histochemical analysis of inflamed tissues, tumor tissues, and other immunohistopathological applications.

In recent years, the possible involvement of CD68 in atherosclerosis was shown in in vitro and in vivo experiments. Yamanouchi et al. observed SR-AI, CD68, and receptor expression for advanced glycation endproducts (RAGE) mRNA in foam cells accumulated in early atherosclerotic lesions.[1] Duan et al. Showed that CD68 ablation enhances atheroma stability and suppresses macrophage apoptosis.[2] Indeed, these observations suggested a likely role of CD68 in foam cell formation. In summary, CD68 is not only a reliable marker of macrophages but also represents the inflammatory level of plaques and correlates with plaque vulnerability, so this study assessed the extent of macrophage expression by measuring CD68 expression levels.

References

[1] Yamanouchi J, Takatori A, Nishida E, Kawamura S, Yoshikawa Y. Expression of lipoprotein receptors in the aortic walls of diabetic APA hamsters. Exp Anim. 2002 Jan;51(1):33-41.

[2] Duan L, Zhao Y, Jia J, Chao T, Wang H, Liang Y, Lou Y, Zheng Q, Wang H. Myeloid-restricted CD68 deficiency attenuates atherosclerosis via inhibition of ROS-MAPK-apoptosis axis. Biochim Biophys Acta Mol Basis Dis. 2023 Jun;1869(5):166698.